# Attitudes towards LGBTQ+ individuals among Thai medical students

**Teeravut Wiwattarangkul[1], Sorawit Wainipitapong[1,2]** *

**1** Department of Psychiatry, Faculty of Medicine, Chulalongkorn University and King Chulalongkorn Memorial Hospital, Bangkok, Thailand, **2** Department of Global Health and Social Medicine, King's College London, United Kingdom and Center of Excellence in Transgender Health (CETH), Faculty of Medicine, Chulalongkorn University, Bangkok, Thailand

\* sorawit.wainipitapong@kcl.ac.uk; sorawit.w@chula.ac.th

## Abstract

### Background

The global population of individuals with gender diversity or LGBTQ+ people is on the rise. However, negative attitudes towards LGBTQ+ individuals persist, even among healthcare professionals, creating barriers to healthcare access. These attitudes are influenced by cultural variations worldwide and necessitate investigation across diverse cultures and settings.

### Objectives

This study aimed to evaluate the attitudes towards LGBTQ+ people and describe associated factors with being LGBTQ+ among Thai medical students.

### Methods

During the 2021 academic year, a survey was conducted at a medical school in Bangkok, Thailand, collecting demographic data and attitudes measured by a standardised Thai questionnaire. Descriptive statistics as well as bivariate and multivariable logistic regression analyses were used to describe characteristics and association.

### Results

A total of 806 medical students participated, with a neutral attitude being the most prevalent (72.2%), followed by a positive attitude (27.2%), and a minority reporting a negative attitude (0.6%). Bivariate and multivariable logistic regression analyses revealed significant associations between positive attitudes and female sexual identity (aOR 2.02, 95%CI 1.45–2.81, p-value < 0.001), having LGBTQ+ family members (aOR 3.57, 95%CI 1.23–10.34, p-value = 0.019), having LGBTQ+ friend (aOR 1.46, 95%CI 1.02–2.11, p-value = 0.040), and coming from areas outside of Bangkok (aOR 1.41, 95%CI 1.01–1.97, p-value = 0.043).

**Data Availability Statement:** All files are available from the Harvard Dataverse: Wainipitapong, Sorawit, 2023, "Attitudes towards LGBTQ+ among

Thai medical students", https://doi.org/10.7910/DVN/ELRWTI, Harvard Dataverse, V1.

**Funding:** The authors received no specific funding for this work.

**Competing interests:** The authors have declared that no competing interests exist.

## Conclusion

Positive attitude towards the LGBTQ+ community are essential for physicians, emphasising the need to study factors that contribute to positive attitudes in order to foster an LGBTQ +-friendly environment for both patients and medical students.

## Background

The number of people identifying as gender diverse or LGBTQ+ is increasing, particularly among younger populations. Research shows that the proportion of LGBTQ+ students has risen from 11.2% in 2015 [1] to 15.6% in 2019 [2]. Despite progress, negative attitudes towards the LGBTQ+ community persist globally, leading to discrimination, sexual violence, employment challenges, and barriers to healthcare access [3]. These negative attitudes pose obstacles to achieving a better quality of life for the LGBTQ+ community [4].

Negative attitudes towards the LGBTQ+ community, also known as 'homonegative attitudes', are prevalent worldwide but can vary in intensity across different locations. These attitudes have detrimental effects at both the individual and societal levels [5, 6]. LGBTQ+ individuals often face barriers in accessing certain services and employment opportunities due to homonegativity, which is a disadvantage not adequately acknowledged by the majority [7].

Homonegativity is associated with various factors such as age, religiosity, masculinity, heterosexuality, low educational level, lower academic performance, high authority, and sanctity [6, 8, 9]. These negative attitudes contribute to difficulties in accessing difficulties in accessing both physical and mental health services for LGBTQ+ individuals, leading to disparities in healthcare [10, 11]. Due to fears of gender discrimination, LGBTQ+ individuals may be hesitant to disclose their orientation or identity when seeking healthcare, highlighting the need for healthcare professionals to develop LGBTQ+ acceptance and attitudes to provide optimal care [12, 13].

Some studies have shown that attitudes towards the LGBTQ+ population are associated with the readiness to care for LGBTQ+ patients [14, 15]. Negative attitudes are often associated with factors such as male gender, religiosity, and the absence of an LGBTQ+ friend [16, 17]. These negative attitudes can be associated with lower acceptance, discrimination, and mental health issues, which, in turn, have been found to be related with medical schools dropout [18]. On the contrary, positive attitudes are correlated with higher levels of LGBTQ+ knowledge and confidence among medical students in providing care to LGBTQ+ patients [14, 15, 19–21].

Several questionnaires have been developed to measure attitudes towards the LGBTQ+ community, including the Attitudes of Heterosexuals Toward Homosexuality (HATH), the Attitudes Toward Lesbians and Gay Men (ATLG), the Multidimensional Scale of Attitudes Toward Lesbians and Gay, and the Lesbian, Gay, Bisexual, and Transgender Development of Clinical Skills Scale (LGBT-DOCSS), which have been validated [22, 23]. However, none of these measures have been translated into Thai. To address this, a national questionnaire with 12 items and a five-point Likert scale has been created in Thailand, specifically designed to fit the psychosocial and cultural context of the country, demonstrating good validity and reliability [24].

Understanding the attitudes of medical students towards LGBTQ+ individuals is crucial for revising an LGBTQ+ inclusive curriculum that prepares students to provide care to LGBTQ + patients and become LGBTQ+-friendly professionals [14, 15, 19–21]. Since attitudes differ globally, it is important to conduct psychosocially and culturally specific studies to investigate attitudes, and previous research has primarily focused on non-Asian contexts. Therefore, the

primary objective of our study was to evaluate the attitudes towards LGBTQ+ people among Thai medical students, taking into account the unique cultural and social factors that may influence their perspectives. Such as an examination is instrumental in evaluating whether negative attitudes exist, as they can subsequently lead to issues like discrimination, mental health problems, and even medical school dropouts. Additionally, we aimed to identify and describe the associated factors related to identifying as LGBTQ+ compared to students who identify as non-LGBTQ+. Considering limited resources and the need for time-effective education, examining these factors allowed us to gain insights into the specific challenges and considerations that need to be addressed in developing an LGBTQ+ inclusive curriculum for medical education in Thailand. We also collected opinions from medical students on LGBTQ + inclusive curriculum to gather student-centred evidence. This approach also enabled us to modify the curriculum based on both student voices and the necessary components related to LGBTQ+ attitudes.

## Materials and methods

During the 2021 academic year, we included all first to sixth-year medical students aged 18 years old or older from the Faculty of Medicine, Chulalongkorn University, Thailand. A total of 806 students participated in the study. These students were in their preclinical year at the faculty and were assigned to four affiliated hospitals based on their chosen programme. We excluded students who were on academic leave or unable to communicate in Thai. Sample size calculation was performed using the Yamane equation [25], determining a minimum of 328 participants. As all eligible students were invited, no additional sampling was required. We obtained informed consent from participants through an online platform. Our study received approval from the Institutional Review Board of the Faculty of Medicine, Chulalongkorn University, Bangkok, Thailand (COE No.002/2022).

We collected demographic data, including sexual and gender identity, sexual orientation, age, academic year, religion and religiosity, family history, hometown, and academic performance. Furthermore, we investigated the presence of LGBTQ+ individuals within participants' families and friends, the inclusion of LGBTQ+-related classes during the curriculum, and perspectives on LGBTQ+ in medical education. To ensure participant confidentiality and mitigate potential discrimination or unwillingness to disclose their identity, our study utilised an anonymous, self-reported questionnaire [6, 8, 9]. This approach aimed to create a safe space for participants to provide honest responses.

### Attitudes towards LGBTQ+ measure

To assess participants' attitudes towards LGBTQ+, we utilised the Thai Attitude towards LGBTQ+ questionnaire. This instrument divides attitudes into three categories: positive, neutral, and negative. The questionnaire consists of 12 items measured on a five-point Likert scale, with response options ranging from 1 (strongly disagree) to 5 (strongly agree). A higher total score indicates a more positive attitude. The content validity and reliability of the questionnaire were previously investigated, demonstrating good psychometric properties with Cronbach's alpha coefficient exceeding 0.75 in each subscale [24]. Attitude scores were categorised as follows: 28 or lower represented a negative attitude, 29–45 indicated a neutral attitude, and scores higher than 45 reflected a positive attitude.

### Opinions regarding LGBTQ+ in medical education

As a standard questionnaire on LGBTQ+ in medical education has not been yet developed, we created a five-item questionnaire to assess opinions on this matter. We employed a 10-point

Likert scale for responses, enabling us to observe the binary direction of agreement or disagreement, and also grade the level of agreement. Each item addressed the adequacy of knowledges regarding LGBTQ+ care acquired through either compulsory or elective modules, the inclusivity of LGBTQ+ studies in the curriculum, the right to dress in alignment with gender identity, and the confidence to provide care for LGBTQ+ patients after graduation.

## Data analysis

Descriptive statistics were used to summarise the demographic data of all participants. Categorical variables were presented as counts and percentage, while continuous variables were reported as means with standard deviation. Chi-square tests, student t-tests, and one-way ANOVA were employed based on the nature of each variable. The association between attitude and related factors was examined through bivariate and multivariable logistic regression analyses. Only variables that exhibited a statistically significant association in the bivariate analysis (p-value < 0.05) or those aligned with theoretical concepts were included in the multivariable analysis. Given the presence of missing data, multiple imputation using the Multiple Imputation by Chained Equations (MICE) approach was utilised, generating ten imputed datasets. Pooled regression estimates from the ten imputed datasets were reported. A p-value of < 0.05 was considered statistically significant. Data analysis was performed using SPSS version 22.0.

## Results

A total of 806 participants out of 1903 medical students (42.4%) from the 2021 academic year completed the questionnaire. The majority of participants identified as cisgender (91.1%) and expressed satisfaction with their gender (82.0%). Male participants slightly outnumbered females, accounting for 52.7% of the respondents. In terms of sexual orientation, heterosexual individuals constituted the largest (76.9%), followed by bisexual/pansexual individuals (11.7%), and homosexual individuals (7.4%). The mean age was 21.2 years old, and the majority attended the university hospital (73.7%). Buddhism was the most prevalent religion among participants (88.6%), although a minority reported being strongly religious (8.6%). Approximately half of the participants resided in Bangkok (51.9%), and currently lived with their parents (56.8%). Only a small proportion reported having LGBTQ+ family members (2.0%), but a majority had at least 1 LGBTQ+ friend (64.4%). **Table 1** presents the demographic data in detail.

Opinions regarding LGBTQ+ in medical education were measured using ten-point Likert scales. The results showed high agreement with the idea that individuals should dress according to their gender identity (mean score of 8.8 ± 1.8), and that LGBTQ+ studies should be incorporated into the curriculum (mean score of 7.9 ± 2.1). However, participants disagreed that LGBTQ+ knowledge was adequately covered in both compulsory courses and elective courses (mean score of 4.9 ± 2.6 and 3.7 ± 2.9, respectively). Topics related to LGBTQ+ knowledge were found to be included in the Reproductive system course during the preclinical year (33.9%) and in Psychiatry courses during both the preclinical and clinical year (33.6%).

### Attitudes towards LGBTQ+ and associated factors

According to the questionnaire results, only five participants (0.6%) demonstrated a negative attitude towards LGBTQ+. The most prevalent attitude among the medical students was a neutral attitude (72.2%). A positive attitude was found in 219 medical students (27.2%), and their total score was significantly higher compared to the non-positive group (composed of those with negative to neutral attitudes) (48.3 ± 2.3 VS 40.1 ± 3.9, p-value < 0.001). This

**Table 1. Demographic data (n = 806).**

| Variables | N (%) or Mean ± S.D. | Variables | N (%) or Mean ± S.D. |
|---|---|---|---|
| Sexual identity | | Gender normativity | |
| Male | 425 (52.7) | Normative | 620 (76.9) |
| Female | 379 (47.0) | LGBTQ+ | 174 (21.6) |
| Prefer not to say | 2 (0.3) | Unspecified | 12 (1.5) |
| Gender identity | | Sexual orientation | |
| Cisgender | 739 (91.7) | Heterosexual | 628 (77.9) |
| Transgender | 9 (1.1) | Homosexual | 60 (7.4) |
| Non-binary | 23 (2.9) | Bisexual/pansexual | 93 (11.5) |
| Questioning | 29 (3.6) | Asexual | 12 (1.5) |
| Prefer not to say | 6 (0.7) | Prefer not to say | 13 (1.6) |
| Age (years) | 21.2 ± 1.7 | Gender satisfaction | |
| Educational year | | Satisfied | 661 (82.0) |
| Preclinical year | 430 (53.3) | Neutral | 142 (17.6) |
| Clinical year | 376 (46.7) | Unsatisfied | 3 (0.4) |
| Education program | | Religion | |
| University hospital | 594 (73.7) | Buddhism | 714 (88.6) |
| Affiliated hospitals | 197 (24.4) | Christianity | 21 (2.6) |
| International program | 15 (1.9) | Islam | 3 (0.4) |
| | | Non-religious | 68 (8.4) |
| Academic performance | | Religiosity | |
| GPAX* | 3.4 ± 0.5 | Self-religiosity | 70 (8.6) |
| GPA** | 3.4 ± 0.5 | Family-religiosity | 44 (5.5) |
| Hometown | | Current living places | |
| Bangkok | 418 (51.9) | With parents | 458 (56.8) |
| Non-Bangkok | 385 (47.8) | With siblings | 47 (5.8) |
| Missing | 3 (0.4) | With friends | 138 (17.1) |
| Parental marital status | | Alone | 163 (20.2) |
| Married/cohabited | 703 (87.2) | | |
| Separation/divorced | 77 (9.6) | | |
| Widowed | 26 (3.2) | | |
| Paternal educational level | | Maternal educational level | |
| Lower than Bachelor | 86 (10.7) | Lower than Bachelor | 79 (9.8) |
| Bachelor or higher | 704 (87.3) | Bachelor or higher | 711 (88.2) |
| Unknown | 16 (2.0) | Unknown | 16 (2.0) |
| Having LGBTQ+ family members | 16 (2.0) | Number of LGBTQ+ friend | |
| | | 0 persons | 283 (35.1) |
| | | 1–5 persons | 409 (50.7) |
| | | >5 persons | 111 (13.7) |
| | | Missing | 3 (0.4) |

*GPAX—Accumulated grade point average

**GPA—Grade point average

difference was consistent across all individual items of the attitude instrument used in this study (24), as indicated in **Table 2**.

**Table 2. Attitude towards LGBTQ+ questionnaire (24) and difference between the positive and non-positive group.**

| Question item | Total (n = 806) | Positive (n = 219) | Non-Positive (n = 587) | P-value |
|---|---|---|---|---|
| | Mean ± S.D. | | | |
| 1 Being LGBTQ+ is a disorder ** | 4.4 ± 1.0 | 4.9 ± 0.7 | 4.3 ± 1.1 | <0.001* |
| 2 Being LGBTQ+ is against nature ** | 4.5 ± 0.9 | 5.0 ± 0.3 | 4.4 ± 1.0 | <0.001* |
| 3 Thai society warmly welcomes LGBTQ+ community | 2.7 ± 1.0 | 2.9 ± 1.1 | 2.6 ± 0.9 | 0.001* |
| 4 The LGBTQ+ community tends to be skillful | 3.0 ± 0.8 | 3.2 ± 0.8 | 3.0 ± 0.8 | 0.001* |
| 5 Being LGBTQ+ is not a choice | 3.1 ± 1.2 | 3.3 ± 1.3 | 3.0 ± 1.1 | <0.001* |
| 6 The LGBTQ+ community is more expressive than straights ** | 3.1 ± 1.1 | 3.6 ± 1.0 | 3.0 ± 1.0 | <0.001* |
| 7 Being LGBTQ+ affects job applications and employment ** | 2.6 ± 1.1 | 3.4 ± 1.3 | 2.4 ± 0.9 | <0.001* |
| 8 The LGBTQ+ community should be refrain from some occupations ** | 4.2 ± 1.2 | 4.9 ± 0.3 | 4.0 ± 1.3 | <0.001* |
| 9 Being LGBTQ+ affects family descent ** | 3.1 ± 1.3 | 4.2 ± 1.0 | 2.7 ± 1.1 | <0.001* |
| 10 The LGBTQ+ community has legal struggles ** | 2.6 ± 1.4 | 3.4 ± 1.6 | 2.3 ± 1.1 | <0.001* |
| 11 LGBTQ+ marriage should be allowed | 4.5 ± 0.9 | 4.8 ± 0.5 | 3.4 ± 0.9 | <0.001* |
| 12 Adoption should be legalised for LGBTQ+ couples | 4.3 ± 0.9 | 4.7 ± 0.7 | 4.2 ± 1.0 | <0.001* |
| **Total score** | 42.3 ± 5.1 | 48.3 ± 2.3 | 40.1 ± 3.9 | <0.001* |

*P-value significant at 0.05

**Negative questions with reverse rating score of agreement

## Bivariate and multivariable logistic regression of positive attitude

To further explore the association between the positive attitude group and the non-positive group, bivariate and multivariable logistic regression analyses were performed and are presented in **Tables 3 and 4**, respectively. In the bivariate logistic regression analysis, several factors were found to be significantly associated with a positive attitude towards LGBTQ+. These factors included female sexual identity, having LGBTQ+ family members and friend, and coming from areas outside of Bangkok. Additionally, student's t-test analysis showed that the total attitude score was significantly higher for those who were female, had LGBTQ+ friend, and came from areas outside of Bangkok. However, no significant difference in attitude was found when comparing those who had LGBTQ+ family members to those who did not.

After adjusting for years of education, religiosity, and academic performance in the multivariable logistic regression analysis, the associations remained significant. Female sexual identity (adjusted odds ratio (aOR) = 1.98, 95%CI 1.42–2.77, p-value < 0.001), having LGBTQ+ family members (aOR = 3.58, 95%CI 1.24–10.37, p-value = 0.02), and having LGBTQ+ friend (aOR = 1.51, 95%CI 1.04–2.20, p-value = 0.03) were all significantly associated with a positive attitude. These findings were consistent when using the Multiple Imputation by Chained Equations (MICE) approach for handling missing data. **Fig 1** displays the total attitude scores, which are positively a correlated with a good attitude.

## LGBTQ+ medical students

Among the total of 174 participants (21.6%) who identified as LGBTQ+, the most common sexual orientation reported was bisexual (39.1%), while lesbian was the least common (4.0%). One participant (0.6%) identified as a crossdresser/androgynous, and nine participants (5.2%) identified as transgender. In terms of disclosure, the majority of LGBTQ+ participants (68.2%) had disclosed their gender identity to their close friends. This was followed by disclosure to siblings, mothers, classmates, and fathers, respectively. Additional data regarding the LGBTQ+ participants is presented in **Table 5**.

**Table 3. Bivariate logistic regression analysis of a positive attitude.**

| Variables | Bivariate model | |
|---|---|---|
| | OR (95% CI) | P-value |
| Having LGBTQ+ family members | 3.54 (1.30–9.62) | 0.013* |
| Female sexual identity | 2.11 (1.53–2.90) | <0.001* |
| Having LGBTQ+ friend | 1.79 (1.27–2.53) | 0.001* |
| Hometown outside of Bangkok | 1.41 (1.03–1.92) | 0.033* |
| Educational program | 1.27 (0.93–1.72) | 0.132 |
| Parental marital status | 1.27 (0.91–1.76) | 0.160 |
| Self-religiosity | 1.17 (0.68–2.00) | 0.578 |
| Paternal educational level | 1.10 (0.68–1.83) | 0.718 |
| Accumulated grade point average | 1.09 (0.79–1.54) | 0.638 |
| Current living places | 1.06 (0.94–1.20) | 0.328 |
| Gender satisfaction | 1.03 (0.70–1.52) | 0.875 |
| Being LGBTQ+ | 1.00 (0.99–1.01) | 0.869 |
| Religion | 0.99 (0.89–1.11) | 0.862 |
| Age | 0.98 (0.90–1.08) | 0.709 |
| Educational year | 0.97 (0.71–1.33) | 0.853 |
| Family-religiosity | 0.92 (0.45–1.85) | 0.810 |
| Maternal educational level | 0.80 (0.48–1.32) | 0.381 |

*P-value significant at 0.05

## LGBTQ+ and non-LGBTQ+ medical students

When comparing medical students who identified as LGBTQ+ with those who did not, several significant differences were observed, as outlined in **Table 6**. LGBTQ+ students tend to be younger (p-value = 0.004) and were more likely to have LGBTQ+ friends (p-value < 0.001) and LGBTQ+ family members (p-value = 0.001). They also had lower levels of paternal and maternal education (p-value = 0.012 and 0.030, respectively). LGBTQ+ students were more commonly in the preclinical year of their medical studies (p-value = 0.013) and were more likely to come from areas outside of Bangkok (p-value = 0.025).

Regarding their opinions on LGBTQ+ in medical education, LGBTQ+ students expressed strong support for the inclusion of LGBTQ+ studies in the curriculum (p-value < 0.001) and for the right of medical students to dress in accordance with their gender identity (p-

**Table 4. Multivariable logistic regression analysis of a positive attitude.**

| Variables | Multivariable model | | Pooled parameter estimates | |
|---|---|---|---|---|
| | Adjusted OR (95% CI) | P-value | Adjusted OR (95% CI) | P-value |
| Having LGBTQ+ family members | 3.57 (1.23–10.34) | 0.019* | 3.55 (2.07–6.11) | 0.019* |
| Female sexual identity | 2.02 (1.45–2.81) | <0.001* | 2.01 (1.46–2.75) | <0.001* |
| Having LGBTQ+ friend | 1.46 (1.02–2.11) | 0.040* | 1.52 (1.06–2.17) | 0.023* |
| Hometown outside of Bangkok | 1.41 (1.01–1.97) | 0.043* | 1.44 (1.03–2.00) | 0.031* |
| Self-religiosity | 1.16 (0.66–2.05) | 0.598 | 1.19 (0.68–2.06) | 0.548 |
| Accumulated grade point average | 1.12 (0.76–1.65) | 0.564 | 1.15 (0.78–1.69) | 0.480 |
| Educational year | 1.00 (0.89–1.12) | 0.967 | 1.01 (0.91–1.13) | 0.840 |

*P-value significant at 0.05

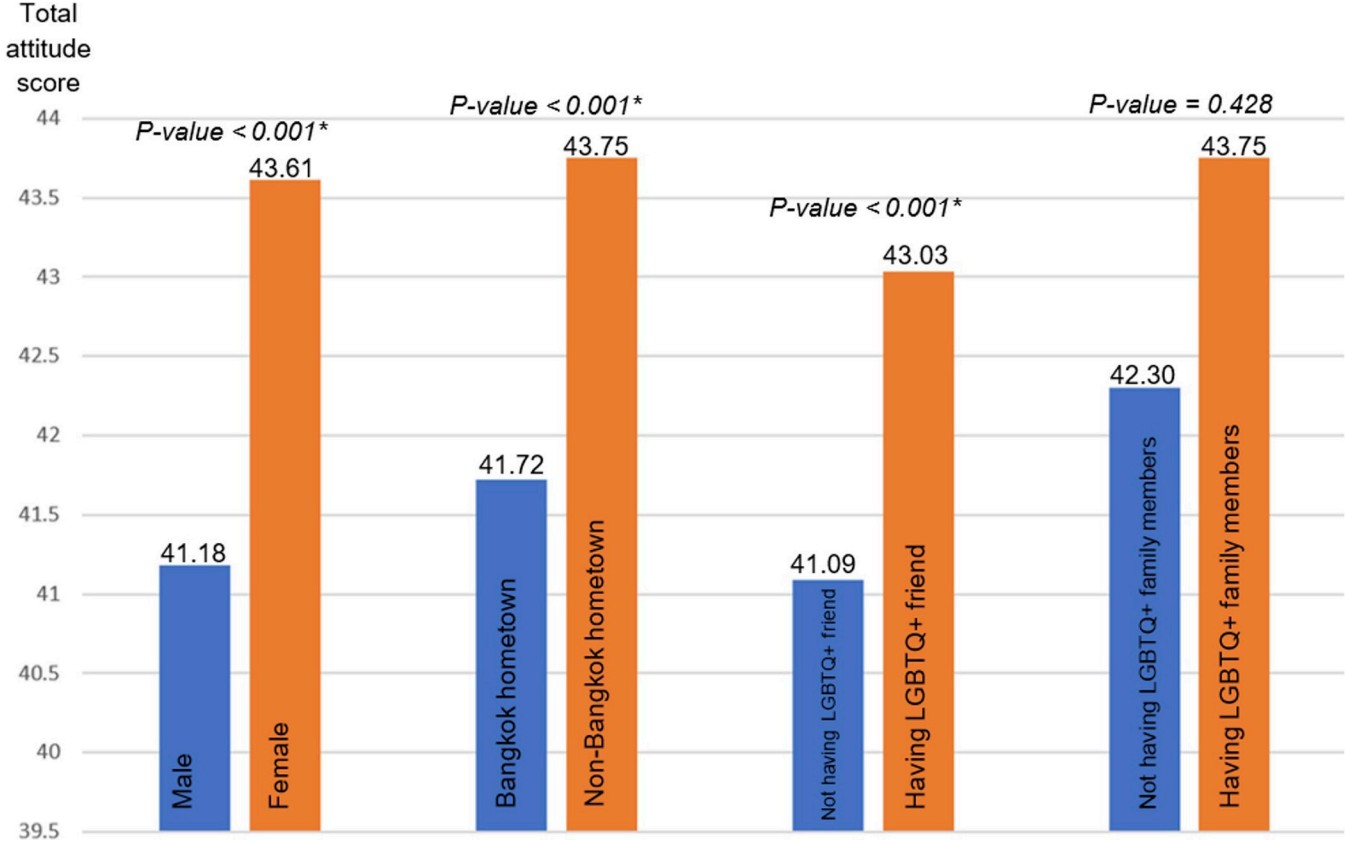

**Fig 1. Total attitude scores and associated factors.**

value = 0.001). Detailed differences between LGBTQ+ and non-LGBTQ+ medical students can be founded in Table 6.

## Discussion

Our study found that a considerable proportion of Thai medical students (27.2%) had a positive attitude towards the LGBTQ+ community, which was significantly higher than the general population (1.0%) [24]. This difference may be attributed to the medical students' greater knowledge about LGBTQ+ issues, which is known to be a protective factor against homonegativity.

**Table 5. LGBTQ+ participants information (n = 174).**

| Variables | N (%) | Variables | N (%) |
|---|---|---|---|
| **Gender diversity** | | **Disclosure** | |
| Lesbian | 7 (4.0) | Father | 35 (20.2) |
| Gay | 28 (16.1) | Mother | 54 (31.2) |
| Bisexual/pansexual | 68 (39.1) | Siblings | 59 (34.1) |
| Transgender | 9 (5.2) | Close friends | 118 (68.2) |
| Questioning | 27 (15.5) | Classmates | 52 (30.1) |
| Asexual | 12 (6.9) | Others | 32 (18.5) |
| Non-binary | 22 (12.6) | | |
| Androgyny | 1 (0.6) | | |

**Table 6. LGBTQ+ students and related factors.**

| Variables | Non LGBTQ+ (n = 620) | LGBTQ+ (N = 174) | P-value |
|---|---|---|---|
| | N (%) or Mean ± S.D. | | |
| Age (years) | 21.3 ± 1.6 | 20.8 ± 1.8 | 0.004* |
| Preclinical year | 315 (50.8%) | 107 (61.5%) | 0.013* |
| Accumulated grade point average | 3.4 ± 0.4 | 3.4 ± 0.5 | 0.367 |
| Hometown outside of Bangkok | 283 (45.9%) | 97 (55.7%) | 0.025* |
| Self-religiosity | 55 (8.9%) | 15 (8.6%) | 1.000 |
| Family-religiosity | 30 (4.8%) | 12 (6.9%) | 0.336 |
| Paternal educational level lower than Bachelor's degree | 56 (9.3%) | 28 (16.2%) | 0.012* |
| Maternal educational level lower than Bachelor's degree | 52 (8.6%) | 25 (14.5%) | 0.030* |
| Have LGBTQ+ friend | 373 (60.4%) | 141 (81.5%) | <0.001* |
| Have LGBT family members | 51 (8.2%) | 30 (17.3%) | 0.001* |
| Obtaining knowledge about LGBTQ+ from compulsory courses | 4.9 ± 2.6 | 4.6 ± 2.8 | 0.074 |
| Obtaining knowledge about LGBTQ+ from elective courses | 3.8 ± 2.9 | 3.1 ± 3.0 | 0.048* |
| Advocating LGBTQ+ study in medical curriculum | 7.7 ± 2.1 | 8.4 ± 1.9 | <0.001* |
| Advocating medical students' right to dress as their gender identity | 8.7 ± 1.8 | 9.2 ± 1.5 | 0.001* |
| Having confidence to care for LGBTQ+ patients | 7.9 ± 2.1 | 8.2 ± 2.2 | 0.113 |
| Good attitude | 171 (27.6%) | 45 (25.9%) | 0.700 |
| Total attitude score | 42.2 ± 5.3 | 42.7 ± 4.1 | 0.247 |

*P-value significant at 0.05

Medical students might have more knowledge about LGBTQ+, which is one protective factor against homonegativity [19, 26]. Additionally, the younger age of medical students in this study was associated with a better attitude, consistent with previous research [6, 24, 27].

The prevalence of homonegativity among Thai medical students was low (0.6%), which contrasts with findings from previous research [28]. These favourable attitudes were protective against adverse consequences, such as discrimination, and we also suspected less resistance from students' view on revising medical curriculum to be more LGBTQ+ inclusive. Some interventions like education can be implemented on those with negative attitudes, helping them to become clinicians for people with diversity. However, it is important to note that the study used a different questionnaire to assess attitudes, limiting direct comparisons with other studies.

The study identified several factors associated with a positive attitude towards LGBTQ + individuals, which were consistent with previous studies [16, 17]. These factors included being female, having LGBTQ+ friend or family members, and coming from areas outside of Bangkok. Males were more likely to have a less positive attitude [24, 26, 29, 30], and having LGBTQ+ friends was associated with a more positive attitude [16, 31, 32], highlighting the importance of personal contact and relationships in shaping attitudes [22]. Furthermore, our study confirmed that individuals with LGBTQ+ family members tend to have more positive attitudes. Growing up with LGBTQ+ individuals in the family can contribute to greater knowledge, understanding, empathy, and ultimately, a positive attitude. This also promotes family acceptance and a better quality of life for LGBTQ+ individuals [33]. These findings align with the understanding that personal relationships and exposure to diverse perspectives are essential for developing positive attitudes towards the LGBTQ+ community. By fostering such environments and promoting inclusivity, society can work towards creating a more accepting and supportive environment for LGBTQ+ individuals.

The finding that students coming from areas outside of Bangkok showed more positive towards LGBTQ+ individuals compared to those from the capital city is interesting. It contrasts with some previous studies suggesting that homonegativity is more common in rural areas [34]. However, it is important to consider the context of Thailand's national surveillance of discrimination, which is understudied and may have influenced the results of previous studies on homonegativity in rural areas [3]. Discrimination is often observed in centralised areas or capital cities like Bangkok, where people with diverse cultures and attitudes gather for occupational opportunities. Discrimination cases have been frequently reported from these areas, which may indicate a higher rate of homonegativity [33]. Living in metropolitan areas can sometimes create a situation where LGBTQ+ individuals feel compelled to disclose their identities, even if they may not want to, due to limited acceptance [35]. These factors may contribute to a poorer attitude towards LGBTQ+ individuals among those from big cities such as Bangkok, providing a possible explanation for the association found in the study.

Contrary to previous research findings, our study did not find a significant association between religiosity and attitude [16, 17, 27]. Almost all of our medical students identified as Buddhists, a religion in which homosexuality is not explicitly addressed [33]. In contrast, some other religions such as Islam, view gender diversity negatively [27, 36, 37].

While previous studies have shown that older individuals tend to exhibit more homonegativity [6, 24, 27], our participants were medical students with ages mostly ranging between 18 to 24 years old and a mean age of 21.2 years. In the Thai medical curriculum, students are required to spend six consecutive years in their medical school before becoming physicians, resulting in little variation in age among our participants. Given the younger and narrower age range of our participants, we did not find age to be associated with attitude, which aligns with the findings from some previous research [16].

Some studies have reported that medical students in higher academic years exhibit more positive attitude due to increased knowledge about LGBTQ+ issues and greater exposure to LGBTQ+ patients [19, 26, 29]. Therefore, it is important to explore whether Thai medical students have sufficient knowledge and experience in providing care for LGBTQ+ patients.

The prevalence of LGBTQ+ individuals among Thai medical students in our study was 21.6%, which was higher compared to reports from other countries [15, 16, 21]. Thailand is known for its warm acceptance of LGBTQ+ people, and the prevalence of LGBTQ+ individuals in Thailand has been increasing [11]. This is in line with the global trend where the number of LGBTQ+ individuals is increasing year by year [1, 2]. We also observed that bisexuality, particularly among females, was the most prevalent sexual orientation, which aligns with global patterns [38].

Our study revealed that LGBTQ+ medical students were more likely to have LGBTQ + friends compared to non-LGBTQ+ medical students. This finding supports the idea that friendships often form around shared interests, providing support and contributing to life satisfaction [39]. The association between having LGBTQ+ friends and being LGBTQ+ was not solely explained by attitudes towards LGBTQ+ individuals, as we did not find any difference in attitude between LGBTQ+ and non-LGBTQ+ medical students. This suggests that the choice of friendships may be influenced by factors beyond attitudes alone. Interestingly, similar patterns of friendship preferences based on gender were observed among both LGBTQ + and non-LGBTQ+ individuals, indicating that non-LGBTQ+ individuals tend to be friends with others of the same gender, which is also reflected within LGBTQ+ friendships [40].

LGBTQ+ medical students were more likely to have other LGBTQ+ individuals in their family. While the role of genetic factors in determining sexual orientation is still inconclusive [41], it can be challenging to differentiate genetic influences from the effects of psychosocial factors. In our study, we found that lower parental educational level was associated with being

LGBTQ+ among medical students. Lower educational attainment often leads to lower socio-economic status [42], which has also been reported to be more common among LGBTQ+ individuals in previous studies [43].

Our study found the association between identifying as LGBTQ+ among medical students and lower parental educational levels. Such correlation may be linked to a subsequent lower socioeconomic status [42, 43]. These findings can be explained through family process models [44] that illustrate the connections between socioeconomic status, which includes parental education and income, and their impact on child behaviour and family interactions. Parental educational level can influence parental beliefs and parenting styles, potentially contributing to issues like domestic violence, verbal aggression, or distant parental attachment. These factors, identified as common experiences in the childhoods of LGBTQ+ individuals [45, 46], can significantly affect children's academic performance, success, behaviour, and even their identity formation [44, 47, 48]. Consequently, parental educational level may play a pivotal role in shaping the experiences and identities of LGBTQ+ medical students. However, it is important to note that our study did not primarily aim to assess this topic, and we needed to control several confounding variables, as mentioned earlier. Furthermore, our observations were limited to a group of medical students based in Bangkok. Future studies conducted in rural areas or non-western countries on these issues are still required, as other hidden factors such as culture or beliefs have yet to be identified. Readers are strongly encouraged to avoid generalisations and apply our findings carefully.

LGBTQ+ students in our study predominantly chose to disclose their sexual orientation or gender identity to their close friends. This may be explained by they perceived more sense of trust, security, and predictable reactions from individuals of similar age and social backgrounds [49, 50]. Conversely, the decision to disclose to family members, especially parents, was less common. This reluctance was potentially influenced by concerns about homonegativity and negative reactions, often driven by fear and rejection [51]. The unpredictable responses from parents after coming out can significantly impact LGBTQ+ individuals' decision [45, 52, 53]. Within the family context, we found that disclosure to siblings was the most prevalent, followed by mothers and fathers. Disclosing to a sibling may be perceived as relatively easier to handle compared to the potential for parental rejection [54, 55]. Siblings often share a closer bond and may be more understanding and accepting of their LGBTQ+ siblings' identities, which could also contribute to a greater likelihood of disclosure to them.

LGBTQ+ medical students were more inclined to advocate for the inclusion of LGBTQ+ studies in the medical curriculum and supporting individuals' right to dress in alignment with their gender identity. These two issues were found to foster LGBTQ+ identity recognition [56]. The incorporation of LGBTQ+ studies into the medical curriculum can play a significant role in enhancing understanding, awareness, and cultural competence among future healthcare professionals regarding LGBTQ+ health issues and the unique needs of LGBTQ+ patients.

The gender identity of medical students can positively impact their work, as adopting a gender-diverse perspective makes it easier to understand others. Acknowledging the challenges faced by LGBTQ+ patients in accessing healthcare [57–59], having gender-diverse physicians can create a more inclusive environment in healthcare clinics. This inclusivity encourages LGBTQ+ patients to feel more relaxed and comfortable seeking medical assistance [60]. However, it is essential to also consider the perspectives of non-LGBTQ+ patients towards LGBTQ+ physicians, especially in various sociocultural contexts. Ultimately, it is crucial to recognise that LGBTQ+ healthcare services should not solely rely on LGBTQ+ physicians. All healthcare professionals should possess the knowledge and skills necessary to understand and support

LGBTQ+ individuals, fostering a patient-centred and welcoming healthcare environment. This understanding should be integrated into the medical curriculum [61].

A higher percentage of LGBTQ+ disclosure may indeed indicate greater acceptance and reduced stigmatisation, which correlates with a more positive attitude towards LGBTQ+ individuals and the increasing number of LGBTQ+ individuals each year [62]. We believe that both disclosure and attitude can influence each other bidirectionally, similar to how this occurs with other stigmatised topics, such as mental health problems in medical schools. A non-negative attitude from both peers and medical teachers can help normalise these issues, promote an LGBTQ+-friendly environment, and ensure that opportunities for LGBTQ+ medical students in further education, including specialty training, are not limited [63].

Our study possessed several noteworthy strengths. Firstly, we gathered a substantial number of medical students, surpassing the participant count of previous studies conducted in Thailand. Despite being a single-university report, our participants came from various regions nationwide, ensuring diversity in their backgrounds. Moreover, we successfully recruited students from all academic years, achieving a satisfactory response rate of 806 out of 1903 medical students (42.4%). By enrolling all participants simultaneously at the end of the last semester, our study captured data that represents reach academic year comprehensively. This temporal alignment allows for a robust portrayal of the attitudes prevalent among Thai medical students.

Some limitations should be acknowledged. The method used to recruit participants relied on voluntary attendance, potentially introducing selection bias as it only represented those who willing chose to participate. Additionally, the absence of international standardised questionnaires measuring attitudes towards LGBTQ+ in Thai language posed a limitation. Instead, we utilised a locally developed measurement tool, which may limit the generalisability of our findings when compared to studies employing different assessment instruments. However, the questionnaire underwent rigorous testing for reliability and validity within the Thai population, demonstrating favourable psychometric properties. Because of the method, all medical students were asked to take part in our study, so results were from only those who willingly attended and might lead to selection bias. International standardized questionnaires measuring attitude toward LGBTQ+ have not been translated into Thai. Using a local measurement tool might limit the generalizability of our study to compare attitude levels measured by different tools in other previous studies. However, the questionnaire was tested for its reliability and validity in the Thai population and showed good psychometric properties. Another limitation is the lack of direct evaluation of participants' knowledge about LGBTQ+ population, despite its known association with attitudes. Future studies should consider assessing knowledge levels to gain a more comprehensive understanding of the factors influencing attitudes. It would also be valuable to investigate the long-term changes in attitude towards LGBTQ+ individuals among medical students throughout their education and the potential impact on LGBTQ+ students themselves. Exploring factors that contribute to promoting LGBTQ+ cultural competency among medical students, who will become future doctors, is also a crucial area for further research.

## Conclusion

Significant proportion of Thai medical students demonstrated neutral to good attitudes toward the LGBTQ+ community. Being female, having LGBTQ+ friend or family members, and coming from areas outside of Bangkok were found to be associated with positive attitudes. Given the growing trend of gender diversity in the global population, fostering positive attitudes toward the LGBTQ+ community is of utmost importance for healthcare providers. Further

research should focus on identifying and addressing the factors that contribute to creating an LGBTQ+-friendly environment for both patients and medical students. By promoting inclusivity and understanding, we can ensure that future physicians are equipped to provide optimal care to all individuals, irrespective of their sexual orientation or gender identity.

## Acknowledgments

We appreciate Dr Yanin Thipakorn Kiatpanabhikul for her kind grammatical review.

## Author Contributions

**Conceptualization:** Teeravut Wiwattarangkul, Sorawit Wainipitapong.

**Data curation:** Sorawit Wainipitapong.

**Formal analysis:** Teeravut Wiwattarangkul, Sorawit Wainipitapong.

**Investigation:** Teeravut Wiwattarangkul, Sorawit Wainipitapong.

**Methodology:** Teeravut Wiwattarangkul, Sorawit Wainipitapong.

**Project administration:** Teeravut Wiwattarangkul.

**Resources:** Teeravut Wiwattarangkul, Sorawit Wainipitapong.

**Writing – original draft:** Teeravut Wiwattarangkul.

**Writing – review & editing:** Sorawit Wainipitapong.

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
