## [Decision Letter · Decision Letter 0]

15 Oct 2023

PONE-D-23-20406Attitudes towards LGBTQ+ individuals among Thai medical studentsPLOS ONE

Dear Dr. Wainipitapong,

Thank you for submitting your manuscript to PLOS ONE. After careful consideration, we feel that it has merit but does not fully meet PLOS ONE’s publication criteria as it currently stands. Therefore, we invite you to submit a revised version of the manuscript that addresses the points raised during the review process.

The reviewers have raised significant methodological issues that need to be elaborated upon by the authors. Some of these aspects, such as the cut-off points of the scales, the sample size determination, and the analytical techniques, are critical factors that directly impact the validity of epidemiological studies. Consequently, I kindly request a careful consideration of these aspects.

We look forward to receiving your revised manuscript.

Kind regards,

Ricardo de Mattos Russo Rafael, Ph.D.

Academic Editor

PLOS ONE

Reviewers' comments:

Reviewer's Responses to Questions

**Comments to the Author**

1. Is the manuscript technically sound, and do the data support the conclusions?

Reviewer #1: Yes

Reviewer #2: Partly

2. Has the statistical analysis been performed appropriately and rigorously? 

Reviewer #1: Yes

Reviewer #2: Yes

3. Have the authors made all data underlying the findings in their manuscript fully available?

Reviewer #1: Yes

Reviewer #2: No

4. Is the manuscript presented in an intelligible fashion and written in standard English?

Reviewer #1: Yes

Reviewer #2: Yes

5. Review Comments to the Author

Reviewer #1: 1. Transfer “Descriptive statistics as well as bivariate and multivariable logistic regression analyses were used to describe characteristics and association” to the methods section in the abstract.

2. In the methodology, include temporal data from the survey in the abstract.

3. Transfer “key points” to the discussion section.

4. Review instrument mentioned in reference 23.

5. Adjust the objectives of the abstract with the objectives described in the text (lines 132-135).

6. Include the total number of participants in the materials section (line 140).

7. Suggest sorting the tables’ data in descending order.

8. Justify the use of ten-point Likert scales (line 198).

9. Review this affirmative “…and the prevalence of LGBTQ+ individuals in Thailand is higher than in most countries.(11)” (line 398).

10. Correct the sentence “This in in line” (line 398).

11. Transfer the confidentiality part (lines 349-352) to the methods.

12. Justify questions from the questionnaire in Table 2. Was it based on an instrument?

13. Consider possible selection bias in the recruitment of participants.

14. Adjust references 11, 31.

Reviewer #2: This study aimed to examine the attitudes of Thai medical students toward LGBTQ+ individuals and describe associated factors with LGBTQ+ status. First, the rationale of conducting this survey needs to be further highlighted. In other words, what implications the results of this study might have regarding medical education needs more discussion.

Methods:

The study ran basic demographic information, compared means using t-test & ANOVA, then ran bivariate & multivariable logistic regression. There are a few questions/comments that the authors can further explain:

1. Why did the author(s) choose to use the Yamane equation to calculate sample size above other methods? (The method assumes the population size is known and uses simple random sample)

2. Why were ten-point Likert scales used for opinions regarding LGBTQ+ in medical education? What were the items and the actual Likert scale?

3. What was the cut-off point/score for the positive and non-positive group? Is it based on the total score of the items? Or individual items? The authors mentioned only five participants demonstrated a negative attitude towards LGBTQ+. Is this based on a single item or the total score?

4. More rationale and discussions should be included on the association between lower parental educational level and LGBTQ+ status among medical students. Same with the disclosure of sexual orientation or gender identity-- more explanations needed.

Also, in the discussion, the authors think that “students spent six consecutive years in their medical school, resulting in little variation in age among our participants”. Hence, why does the author find “age not to be associated with attitude is ‘interesting’” as the mean age of the participants is about 21-22, and is considered ‘young’?

The authors can further discuss how gender identity might affect students’ professional identity and the further impacts these have on the formation of the learning climate (such as hidden curriculum and health care). How might the higher percentage of LGBTQ+ disclosure among students affect the peer students’ attitude can be also discussed. Fostering attitude is pivotal, but to further promote inclusivity and health equity needs further strategies and education. In other words, if the questionnaire is conducted with the general public will there be any differences or similarities as with medical students?

6. PLOS authors have the option to publish the peer review history of their article (what does this mean?). If published, this will include your full peer review and any attached files.

Reviewer #1: **Yes: **Davi Depret

Reviewer #2: No

---

## [Author Response · Author response to Decision Letter 0]

27 Oct 2023

Reviewer #1: Thank you for dedicating your time and providing valuable feedback on our manuscript. We sincerely appreciate all your suggestions, and we have made the following adjustments based on your advice.

Comment 1. Transfer “Descriptive statistics as well as bivariate and multivariable logistic regression analyses were used to describe characteristics and association” to the methods section in the abstract.

Response: Thank you very much. We have now transferred such sentence to the methods section in the abstract (Page 3, Line 70-71).

Comment 2. In the methodology, include temporal data from the survey in the abstract.

Response: This survey was conducted during the 2021 academic year and we have now included the temporal data in the methods section of our revised abstract (Page 3, Line 68).

Comment 3. Transfer “key points” to the discussion section.

Response: Thank you. We have now incorporated all key points in the abstract. The known points have been addressed in the background section (Page 3, Line 63-65), while the results section has encompassed the points added by this study (Page 3, Line 72-78).

Comment 4. Review instrument mentioned in reference 23.

Response: We used the questionnaire developed by Pewnil 2016, which is described in a subsection of the method section. The subheading has been revised to ‘Attitude towards LGBTQ+ measure’ for its conciseness (Page 8, Line 199). Furthermore, we have provided additional details on the development and psychometric properties of this instrument with following sentence: the content validity and reliability of the questionnaire were previously investigated, demonstrating good psychometric properties with Cronbach’s alpha coefficient exceeding 0.75 in each subscale (Page 8, Line 205-206). 

Comment 5. Adjust the objectives of the abstract with the objectives described in the text (lines 132-135).

Response: Thank you for bringing up this point. We have addressed both the primary and secondary objectives of our study in the revised abstract, and the text has been adjusted in accordance with the manuscript as following sentence: this study aimed to evaluate the attitudes towards LGBTQ+ people and describe associated factors with being LGBTQ+ among Thai medical students. (Page 3, Line 66-67).

Comment 6. Include the total number of participants in the materials section (line 140).

Response: Thank you. In the materials section, we have now included your suggested point with following sentence: A total of 806 students participated in the study (Page 7, Line 178-179).

Comment 7. Suggest sorting the tables’ data in descending order.

Response: Thank you very much for this suggestion. We have now sorted two logistic regression tables (Table 3 and 4) in a descending order per your suggestion (Page 13, Line 287 and Page 15, Line 481). Meanwhile, we decided to remain the order within the rest tables for the enhancement of the readability of our readers.

Comment 8. Justify the use of ten-point Likert scales (line 198).

Response: Thank you for this point. As a standard questionnaire on LGBTQ+ in medical education has not been yet developed, we created a five-item questionnaire to assess opinions on this matter. We employed a 10-point Likert scale for responses, enabling us to observe the binary direction of agreement or disagreement, and also grade the level of agreement. Each item addressed the adequacy of knowledges regarding LGBTQ+ care acquired through either compulsory or elective modules, the inclusivity of LGBTQ+ studies in the curriculum, the right to dress in alignment with gender identity, and the confidence to provide care for LGBTQ+ patients after graduation. We have now included the above explanation to justify the use of such scale in the materials section (Page 9, Line 213-220).

Comment 9. Review this affirmative “…and the prevalence of LGBTQ+ individuals in Thailand is higher than in most countries.(11)” (line 398).

Response: Thank you for your keen suggestion. We have now revised this statement for better congruent with previous research cited with following sentence: Thailand is known for its warm acceptance of LGBTQ+ people, and the prevalence of LGBTQ+ individuals in Thailand has been increasing (Page 20, Line 628-630). 

Comment 10. Correct the sentence “This in in line” (line 398).

Response: We have now rectified this sentence using ‘this is in line’ (Page 20, Line 630) 

Comment 11. Transfer the confidentiality part (lines 349-352) to the methods.

Response: Thank you. The confidentiality part has now been transferred to the methods section (Page 8, Line 194-197).

Comment 12. Justify questions from the questionnaire in Table 2. Was it based on an instrument?

Response: Thank you for raising this concern. All items were from the instrument developed to investigate attitude towards LGBTQ+ as described in the methods section. We have now mentioned about this more clearly in the subheading of ‘Attitudes towards LGBTQ+ and associated factors’ in the result section (Page 12, Line 266) as well as the legend of Table 2. (Page 12, Line 269) 

Comment 13. Consider possible selection bias in the recruitment of participants.

Response: Thank you for this immensely helpful advice. We had strong concerns about selection bias due to our study design. To address this, we have noted this issue in the in the limitation paragraph at the end of the discussion section. (Page 24, Line 746-748) 

Comment 14. Adjust references 11, 31.

Response: Thank you very much. We have adjusted these references as per your suggestions. (Page 28, Line 826-830 and Page 31, Line 905-907)

Once again, we sincerely appreciate your time and effort in reviewing our manuscript. We believe these revisions have strengthened the quality and clarity of our work.

Reviewer #2: Thank you for dedicating your time and providing valuable feedback on our manuscript. We sincerely appreciate all your suggestions, and we have made the following adjustments based on your advice.

Comment 1. First, the rationale of conducting this survey needs to be further highlighted. In other words, what implications the results of this study might have regarding medical education needs more discussion.

Response: We greatly appreciate this important point. Examining the attitudes of medical students towards LGBTQ+ individuals is of paramount importance. Such as an examination is instrumental in evaluating whether negative attitudes exist, as they can subsequently lead to issues like discrimination, mental health problems, and even medical school dropouts. Equally significant is its role in shaping a more inclusive LGBTQ+ curriculum based on their voices. This, in turn, better prepares students to offer care to LGBTQ+ patients and become LGBTQ+-friendly professionals. We have now incorporated this on the rationale of our study in the introduction section (Page 6, Line 143-174), as well as in the discussion part (Page 23, Line 712-722).

Comment 2. Why did the author(s) choose to use the Yamane equation to calculate sample size above other methods? (The method assumes the population size is known and uses simple random sample)

Response: Thank you. The Yamane formula is a commonly used method for survey sample size determination. It is favoured for its simplicity and applicability to large population sizes. One notable advantage is that it doesn’t necessitate an estimated proportion of the desired characteristics, which is particularly useful for us as these proportions are yet to be investigated. After consulting with our statistical advisor, we were advised to use this equation to calculate the minimum number of participants needed for our survey.

Comment 3. Why were ten-point Likert scales used for opinions regarding LGBTQ+ in medical education? What were the items and the actual Likert scale?

Response: Thank you for this point. As a standard questionnaire on LGBTQ+ in medical education has not been yet developed, we created a five-item questionnaire to assess opinions on this matter. We employed a 10-point Likert scale for responses, enabling us to observe the binary direction of agreement or disagreement, and also grade the level of agreement. Each item addressed the adequacy of knowledges regarding LGBTQ+ care acquired through either compulsory or elective modules, the inclusivity of LGBTQ+ studies in the curriculum, the right to dress in alignment with gender identity, and the confidence to provide care for LGBTQ+ patients after graduation. We have now included the above explanation to justify the use of such scale in the materials section (Page 9, Line 213-220).

Comment 4. What was the cut-off point/score for the positive and non-positive group? Is it based on the total score of the items? Or individual items? The authors mentioned only five participants demonstrated a negative attitude towards LGBTQ+. Is this based on a single item or the total score?

Response: Thank you for pointing that out. The cut-off points are based on the total score: 28 or lower represented a negative attitude, 29-45 indicated a neutral attitude, and scores higher than 45 reflected a positive attitude. The cut-off score of the questionnaire has been explained in the material section. (Page 8, Line 199-208).

Comment 5. More rationale and discussions should be included on the association between lower parental educational level and LGBTQ+ status among medical students.

Response: Thank you for this point. Our study found the association between lower parental educational level and LGBTQ+ medical students, which might be explained by the Dubow’s family process model. However, the generalisation should be strongly avoided as our study didn’t primarily aim to assess this issue and several confounders needed to be controlled. We have now additionally dedicated a paragraph discussing on this topic: Our study found the association between identifying as LGBTQ+ among medical students and lower parental educational levels. Such correlation may be linked to a subsequent lower socioeconomic status (42,43). These findings can be explained through family process models (44) that illustrate the connections between socioeconomic status, which includes parental education and income, and their impact on child behaviour and family interactions. Parental educational level can influence parental beliefs and parenting styles, potentially contributing to issues like domestic violence, verbal aggression, or distant parental attachment. These factors, identified as common experiences in the childhoods of LGBTQ+ individuals (45,46), can significantly affect children's academic performance, success, behaviour, and even their identity formation (44,47,48). Consequently, parental educational level may play a pivotal role in shaping the experiences and identities of LGBTQ+ medical students. However, it is important to note that our study did not primarily aim to assess this topic, and we needed to control several confounding variables, as mentioned earlier. Furthermore, our observations were limited to a group of medical students based in Bangkok. Future studies conducted in rural areas or non-western countries on these issues are still required, as other hidden factors such as culture or beliefs have yet to be identified. Readers are strongly encouraged to avoid generalisations and apply our findings carefully. (Page 21 Line 664 - Page 22 Line 681).

Comment 6. Same with the disclosure of sexual orientation or gender identity-- more explanations needed.

Response: Thank you for your comments. We have now provided more explanations on the disclosure issue with following sentences: LGBTQ+ students in our study predominantly chose to disclose their sexual orientation or gender identity to their close friends. This may be explained by they perceived more sense of trust, security, and predictable reactions from individuals of similar age and social backgrounds (49,50). Conversely, the decision to disclose to family members, especially parents, was less common. This reluctance was potentially influenced by concerns about homonegativity and negative reactions, often driven by fear and rejection (51). The unpredictable responses from parents after coming out can significantly impact LGBTQ+ individuals’ decision (45,52,53). Within the family context, we found that disclosure to siblings was the most prevalent, followed by mothers and fathers. Disclosing to a sibling may be perceived as relatively easier to handle compared to the potential for parental rejection (54,55). Siblings often share a closer bond and may be more understanding and accepting of their LGBTQ+ siblings’ identities, which could also contribute to a greater likelihood of disclosure to them. (Page 22, Line 683-694).

Comment 7. Also, in the discussion, the authors think that “students spent six consecutive years in their medical school, resulting in little variation in age among our participants”. Hence, why does the author find “age not to be associated with attitude is ‘interesting’” as the mean age of the participants is about 21-22, and is considered ‘young’?

Response: Thank you for this crucial point. Our study explored attitude among medical students with age mostly ranged between 18 to 24 years old. Due to this limited range of age, we found no association between attitude and age like previous research done in older population. We have now revised for the clarification with following sentences: our participants were medical students with ages mostly ranging between 18 to 24 years old and a mean age of 21.2 years. In the Thai medical curriculum, students are required to spend six consecutive years in their medical school before becoming physicians, resulting in little variation in age among our participants. Given the younger and narrower age range of our participants, we did not find age to be associated with attitude (Page 19, Line 615 – Page 20, Line 619).

Comment 8. The authors can further discuss how gender identity might affect students’ professional identity and the further impacts these have on the formation of the learning climate (such as hidden curriculum and health care).

Response: Thank you very much. We have now furthered discussed on this issue according to your suggestion: The gender identity of medical students can positively impact their work, as adopting a gender-diverse perspective makes it easier to understand others. Acknowledging the challenges faced by LGBTQ+ patients in accessing healthcare (57–59), having gender-diverse physicians can create a more inclusive environment in healthcare clinics. This inclusivity encourages LGBTQ+ patients to feel more relaxed and comfortable seeking medical assistance (60). However, it is essential to also consider the perspectives of non-LGBTQ+ patients towards LGBTQ+ physicians, especially in various sociocultural contexts. Ultimately, it is crucial to recognise that LGBTQ+ healthcare services should not solely rely on LGBTQ+ physicians. All healthcare professionals should possess the knowledge and skills necessary to understand and support LGBTQ+ individuals, fostering a patient-centred and welcoming healthcare environment. This understanding should be integrated into the medical curriculum (61). (Page 23, Line 712-722).

Comment 9. How might the higher percentage of LGBTQ+ disclosure among students affect the peer students’ attitude can be also discussed.

Response: Thank you for highlighting this interesting point. A higher percentage of LGBTQ+ disclosure may indicate higher acceptance and reduced stigmatisation, which correlates with a less negative attitude towards LGBTQ+ individuals. We have been provided discussion on this point with following sentences: A higher percentage of LGBTQ+ disclosure may indeed indicate greater acceptance and reduced stigmatisation, which correlates with a more positive attitude towards LGBTQ+ individuals and the increasing number of LGBTQ+ individuals each year (62). We believe that both disclosure and attitude can influence each other bidirectionally, similar to how this occurs with other stigmatised topics, such as mental health problems in medical schools. A non-negative attitude from both peers and medical teachers can help normalise these issues, promote an LGBTQ+-friendly environment, and ensure that opportunities for LGBTQ+ medical students in further education, including specialty training, are not limited (63) (Page 23, Line 724 – Page 24, Line 735).

Comment 10. Fostering attitude is pivotal, but to further promote inclusivity and health equity needs further strategies and education. In other words, if the questionnaire is conducted with the general public will there be any differences or similarities as with medical students?

Response: Thank you very much for this interesting points. Prior to our study, this questionnaire was surveyed in 1313 Thai adults living in the central region of Thailand and a positive attitude was reported as only 1%. Several factors, such as more knowledge about LGBTQ+ and younger age, have been discussed to explain our findings and addressed in the discussion section (Page 17, Line 565-572). We have now also addressed the promotion of inclusivity and stigma reduction in medical schools in our revision (Page 23, Line 724 – Page 24, Line 735).

Once again, we sincerely appreciate your time and effort in reviewing our manuscript. We believe these revisions have strengthened the quality and clarity of our work.

---

## [Decision Letter · Decision Letter 1]

4 Dec 2023

Attitudes towards LGBTQ+ individuals among Thai medical students

PONE-D-23-20406R1

Dear Dr. Wainipitapong,

We’re pleased to inform you that your manuscript has been judged scientifically suitable for publication and will be formally accepted for publication once it meets all outstanding technical requirements.

Kind regards,

Ricardo de Mattos Russo Rafael, Ph.D.

Academic Editor

PLOS ONE

Reviewers' comments:

Reviewer's Responses to Questions

**Comments to the Author**

1. If the authors have adequately addressed your comments raised in a previous round of review and you feel that this manuscript is now acceptable for publication, you may indicate that here to bypass the “Comments to the Author” section, enter your conflict of interest statement in the “Confidential to Editor” section, and submit your "Accept" recommendation.

Reviewer #1: All comments have been addressed

2. Is the manuscript technically sound, and do the data support the conclusions?

Reviewer #1: Yes

3. Has the statistical analysis been performed appropriately and rigorously? 

Reviewer #1: Yes

4. Have the authors made all data underlying the findings in their manuscript fully available?

Reviewer #1: Yes

5. Is the manuscript presented in an intelligible fashion and written in standard English?

Reviewer #1: Yes

6. Review Comments to the Author

Reviewer #1: The article "Attitudes towards LGBTQ+ individuals among Thai medical students" has increased the suggestions made by the reviewer and complies with the parameters of the Plosone edition.

7. PLOS authors have the option to publish the peer review history of their article (what does this mean?). If published, this will include your full peer review and any attached files.

Reviewer #1: **Yes: **Davi Depret

---

## [Editor Report · Acceptance letter]

6 Dec 2023

PONE-D-23-20406R1 

Attitudes towards LGBTQ+ individuals among Thai medical students 

Dear Dr. Wainipitapong:

I'm pleased to inform you that your manuscript has been deemed suitable for publication in PLOS ONE. Congratulations! Your manuscript is now with our production department. 

Kind regards, 

on behalf of

Dr. Ricardo de Mattos Russo Rafael 

Academic Editor

PLOS ONE